# Adjuvant Targeted Therapy in Solid Cancers: Pioneers and New Glories

**DOI:** 10.3390/jpm13101427

**Published:** 2023-09-22

**Authors:** Marco Sposito, Lorenzo Belluomini, Letizia Pontolillo, Daniela Tregnago, Ilaria Trestini, Jessica Insolda, Alice Avancini, Michele Milella, Emilio Bria, Luisa Carbognin, Sara Pilotto

**Affiliations:** 1Section of Innovation Biomedicine—Oncology Area, Department of Engineering for Innovation Medicine (DIMI), University of Verona and University and Hospital Trust (AOUI) of Verona, 37134 Verona, Italy; marcosposito91@gmail.com (M.S.); lorenzo.belluomini08@gmail.com (L.B.); daniela.tregnago2@aovr.veneto.it (D.T.); ilariatrestini92@gmail.com (I.T.); jessica.insolda@aovr.veneto.it (J.I.); alice.avancini@univr.it (A.A.); michele.milella@univr.it (M.M.); 2Comprehensive Cancer Center, Fondazione Policlinico Universitario Agostino Gemelli IRCCS, 00168 Roma, Italy; letiziapontolillo@gmail.com (L.P.); emilio.bria@unicatt.it (E.B.); 3Medical Oncology, Department of Translational Medicine and Surgery, Università Cattolica del Sacro Cuore, 00168 Roma, Italy; 4Gynecology Oncology, Fondazione Policlinico Universitario Agostino Gemelli IRCSS, 00168 Roma, Italy; luisa.carbognin@guest.policlinicogemelli.it

**Keywords:** melanoma, GIST, NSCLC, breast cancer, targeted therapy, adjuvant setting

## Abstract

Targeted therapy (TT) has revolutionized cancer treatment, successfully applied in various settings. Adjuvant TT in resected early-stage gastrointestinal stromal tumors (GIST), melanoma, non-small cell lung cancer (NSCLC), and breast cancer has led to practice-changing achievements. In particular, standard treatments include BRAF inhibitors for melanoma, osimertinib for NSCLC, hormone therapy or HER2 TT for breast cancer, and imatinib for GIST. Despite the undeniable benefit derived from adjuvant TT, the optimal duration of TT and the appropriate managing of the relapse remain open questions. Furthermore, neoadjuvant TT is emerging as valuable, particularly in breast cancer, and ongoing studies evaluate TT in the perioperative setting for early-stage NSCLC. In this review, we aim to collect and describe the large amount of data available in the literature about adjuvant TT across different histologies, focusing on epidemiology, major advances, and future directions.

## 1. Introduction

In the last few decades, targeted therapy (TT) has completely revolutionized the cancer treatment landscape. Starting with the introduction of imatinib for advanced gastrointestinal stromal tumors (GIST) and hormone therapy for breast cancer, the identification of molecular targets for novel therapy has been the objective of the scientific community, in the light of precision oncology, which represents an established reality in the treatment of advanced cancers.

On the wave of the promising results obtained in the treatment of metastatic disease, an increasing number of studies are testing TT in the adjuvant setting, anticipating this approach to the early stages. Across some tumors, adjuvant TT is already a reality supported by solid data; in others, such as non-small cell lung cancer (NSCLC), substantial improvements have been recently achieved [1].

Imatinib was the first targeted therapy to be approved in the adjuvant setting. In the early 2000s, two trials, the Z9001 and Scandinavian Sarcoma Group trials, demonstrated the efficacy of adjuvant therapy with imatinib in resected high-risk GIST [2,3]. Based on these results, imatinib was approved by the European Medicines Agency (EMA) in the adjuvant setting for one year and three years of treatment. Based on the COMBI-AD trial [4], in 2018, dabrafenib and trametinib have been approved as adjuvant combination therapy for patients affected by stage III BRAFV600-mutant melanoma patients. In the landscape of breast cancer treatment, adjuvant TT stands as a cornerstone. However, there are several recent or ongoing studies aimed at further improving the outcome in this early disease setting.

Finally, resected lung cancer patients, in stage IB-IIIA NSCLC with EGFR exon 19 deletions or exon 21 L858R substitution mutations, can now benefit from adjuvant osimertinib after EMA approval in May 2021, based on the results of the ADAURA trial [5] (Figure 1).

In most trials leading to the approval of targeted therapies in the adjuvant setting, disease-free survival (DFS) has been the primary endpoint, while overall survival (OS) has been the secondary endpoint. While the first measures the time from the start of treatment until the recurrence of the disease, the second measures the time from the start of treatment until death from any cause.

OS has long been regarded as the most clinically significant outcome in clinical trials, offering a comprehensive evaluation of a treatment’s overall effectiveness. Nevertheless, in the context of early-stage cancer treatment trials, particularly when assessing perioperative or adjuvant therapies, OS can be influenced by the complexities of subsequent multiline therapies post-tumor recurrence. Consequently, alternative endpoints like DFS and other endpoints (such as pathological response, invasive disease-free survival, and event-free survival) are considered.

Here we discuss the past, the present, and the possible future directions of adjuvant targeted therapy in melanoma, GIST, NSCLC, and breast cancer.

## 2. Melanoma

### 2.1. Epidemiology and Prognosis

In the last few decades, the incidence of melanoma has grown steadily, with about 99,780 estimated new cases in 2022 in the United States, accounting for over 90% of skin cancer deaths due to its propensity to early systemic dissemination. Early stages are the most frequent clinical presentation with 78% of cases diagnosed in stages I and II (defined by the absence of nodal metastasis) and an excellent post-surgical excision 5-year survival rate that ranges from 98% (stage I) to 90% (stage II) [6]. However, the melanoma-specific 5-year survival rate ranges from 93% in stage IIIA disease (1–3 clinically occult, tumor-involved lymph nodes (N1a or N2a) and T1a, T1b, or T2a primaries) to 32% for those with stage IIID disease (patients with a thick and ulcerated primary (T4b), and either ≥4 tumor-involved regional nodes (N3a or N3b) or ≥2 tumor-involved nodes and evidence of microsatellite, satellite, or in-transit metastases (N3c)) [7]. Notably, approximately 40% of newly diagnosed melanomas harbor BRAF oncogenic mutation (V600E and V600K as the most frequent) [8].

Treatments targeting BRAF mutations have been translated from the metastatic into the adjuvant setting in stage III, achieving a significant benefit in terms of both relapse-free survival (RFS) and overall survival (OS) [4].

The main clinical trials exploring TT in the adjuvant setting are reported in Table 1, while selected ongoing trials are reported in Table 2.

### 2.2. Major Advances in the Adjuvant Setting

Until 2012, interferon alfa (IFN-α) represented the only effective (and approved) adjuvant treatment available [12]. In 2015, the EORTC 18071 trial compared ipilimumab (an anti-CTLA4 antibody) and placebo in patients with stage III melanoma, demonstrating the benefit of ipilimumab in RFS, OS, and distant metastasis-free survival (DMFS) [13]. However, considering the concerning immune-related side effects reported with ipilimumab, the use of this agent in the adjuvant setting was not approved by the EMA, prioritizing anti-PD-1 or anti-BRAF/MEK treatment.

Different approaches targeting BRAF mutations have been explored. In the BRIM-8 trial, patients with radically resected melanoma and BRAF V600 mutation were randomized to receive double-blind vemurafenib (BRAF inhibitor) 960 mg ×2 daily or placebo for 12 months. The study enrolled patients in two different cohorts (cohort 1, stage IIC, IIIA, IIIB; cohort 2, stage IIIC). The primary endpoint was disease-free survival (DFS). The study showed no statistically significant difference in the hazard ratio (HR) for cohort 2 (HR = 0.80, 95% confidence interval (CI) 0.54–1.18) with a median DFS of 23.1 months in the vemurafenib group and 15.4 months in the placebo group. In cohort 1, the median DFS was not reached in the vemurafenib group, while it was 36.9 months in the placebo group (HR = 0.54, 95% CI 0.37–0.78, *p* = 0.0010) [9] (Table 1).

Currently, anti-PD-1 or BRAF-directed adjuvant treatments are indicated in melanoma patients with stage IIIA (with sentinel lymph node metastasis >1 mm), IIIB, IIIC, and IIID.

Regarding BRAF targeting, the efficacy of the combination of dabrafenib (BRAF inhibitor) and trametinib (MEK inhibitor) was demonstrated in the randomized, double-blind, phase III COMBI-AD study [4]. This study enrolled 870 stage IIIA, IIIB, or IIIC (according to AJCC 7) patients, harboring the BRAF V600E/K mutation. Patients were randomized to receive dabrafenib 150 mg ×2 day + trametinib 2 mg daily for 12 months vs. placebo, after radical surgery. The risk of disease recurrence was reduced by 53% in the TT arm compared to the control group (HR = 0.47, 95% CI 0.39–0.58, *p* < 0.001). The estimated 3-year OS was 86% vs. 77% in the two groups, with an HR = 0.57 (95% CI 0.42–0.79, *p* = 0.0006) in favor of the treatment arm, but this result cannot be considered significant considering the statistical limitations imposed by the interim analysis [4]. At ASCO 2020, the update with 5-year data from the COMBI-AD study was presented. At 5 years, RFS was 52% vs. 36% (HR = 0.51, 95% CI 0.42–0.61) in the dabrafenib plus trametinib vs. placebo arm, respectively [14]. The subgroup analysis and a subsequent post hoc analysis, conducted by re-staging patients enrolled according to the eighth version of the AJCC, showed no clinically significant differences [15]. Based on this evidence, the FDA and EMA approved the combination of dabrafenib–trametinib for resected stage III BRAFV600-mutant melanoma patients (Table 1).

### 2.3. Future Directions

The encouraging results obtained in the COMBI-AD trial led to the evaluation of novel approaches involving BRAF TT in the adjuvant setting. Ongoing studies include the Columbus-AD study (NCT05270044) evaluating the efficacy and safety of 12 months of BRAF (encorafenib) and MEK (binimetinib) inhibitors versus placebo in resected stage IIA/B/C melanoma patients. Furthermore, TT is also under evaluation in a combined neoadjuvant/adjuvant strategy (Table 2).

A perioperative combination of encorafenib plus binimetinib will be compared to standard adjuvant therapy in the phase II PREMIUM trial in BRAF V600E mutant patients in stage III (B/C/D) or oligometastatic resectable stage IV melanoma (NCT05097378). To date, a phase II study randomized 21 patients to receive the neoadjuvant/adjuvant treatment with dabrafenib and trametinib or the standard surgery ± adjuvant therapy. At the interim analysis, the perioperative strategy demonstrated longer event-free survival with 19.7 months, versus 2.9 months for standard care (HR = 0.016, 95% CI 0.00012–0.14, *p* < 0.0001) [10]. NeoCombi was a phase II study evaluating the efficacy of dabrafenib plus trametinib for 12 weeks before surgery, followed by 40 weeks of adjuvant therapy. The rate of pathological complete response (pCR) and the proportion of patients achieving a response at week 12 were the primary endpoints. At a median follow-up of 27 months, RECIST response was achieved by 86% of patients (46% complete responses and 40% partial response). After surgery, all patients achieved a pathologic response (49% pCR and 51% non-complete pCR) [11] (Table 1).

An interesting combination of immunotherapy and anti-BRAF/MEK TKIs will be evaluated in the NEO-TIM trial, a randomized phase II study, aiming to assess the perioperative approach of the combination of vemurafenib, cobimetinib (MEK inhibitor), and atezolizumab (anti-PD-L1) in stage IIIB/C/D and oligometastatic stage IV patients (BRAF wild-type and mutated) (NCT04722575).

## 3. Gastrointestinal Stromal Tumor (GIST)

### 3.1. Epidemiology and Prognosis

GIST accounts for less than 1% of all malignancies, but they are the most common mesenchymal tumors of the gastrointestinal tract [16].

The incidence is 1.5 cases/100,000/year. The most frequent sites of onset are in order: stomach (50%), small intestine (25%), esophagus (<5%), rectum (5%), and extra-intestinal localizations (<5%) [17]. The 5-year survival rates for GIST may vary from 95% for patients with localized disease to 52% for patients with advanced disease.

The histopathological diagnosis of GIST should include the mitotic index, tumor size, and consideration of the tumor’s anatomical location. Notably, the integrity of the tumor capsule represents a crucial prognostic factor in terms of risk of recurrence. Different approaches have been proposed to assess the risk of recurrence based on these prognostic factors, including nomograms and classifications incorporating mitotic index and tumor diameter.

About 85% of sporadic GISTs are characterized by the presence of mutations in tyrosine kinase receptors KIT and PDGFR-α genes, responsible for constitutive receptor activation and downstream proliferative signaling cascade [18]. Notably, KIT and PDGFR-α mutations are mutually exclusive. About 10% of adult GISTs and 85% of pediatric GISTs are associated with genetic syndromes, without KIT and PDGFR-α mutations [19]. The mutational analysis of the KIT and PDGFR genes is both a prognostic and predictive parameter of response to tyrosine kinase receptor inhibitors (TKIs) [16].

The main clinical trials exploring TT in the adjuvant setting are reported in Table 3, while selected ongoing trials are reported in Table 4.

### 3.2. Major Advances in the Adjuvant Setting

Initially, conventional chemotherapy and radiation therapy have been of limited value as adjuvant treatments in radically resected GIST [21]. In 1998, Hirota et al., described for the first time, in five patients with GIST, the presence of an exon 11 mutation in the c-kit proto-oncogene [22]. Simultaneously, a molecule developed for the treatment of chronic myeloid leukemia, imatinib, demonstrated activity against aberrant gene products of KIT and PDGFR-α [21,23,24]. Based on the impressive efficacy in the first patient with metastatic GIST treated with imatinib [25], several trials, subsequently, confirmed the efficacy of imatinib in about 85% of patients with advanced GIST [26], becoming the first TKI approved in the treatment of solid tumors.

Surgical resection represents the primary choice in the treatment of localized GIST. After complete resection, adjuvant therapy with imatinib for 3 years is the standard of care in patients with a high risk of recurrence (risk assessed by mitotic index, neoplasm size, anatomical site, and rupture of the tumor at the time of surgery) and KIT or PDGFR-α sensitive mutation [27]. This type of approach was based on the results of three different trials.

The US American College of Surgeons Oncology Group (ACOSOG) Z9001 trial compared adjuvant treatment with imatinib 400 mg for one year vs. placebo after macroscopically complete surgical resection in patients with localized GIST with a diameter greater than 3 cm. The study demonstrated an advantage in high-risk patients treated with imatinib in terms of RFS, while no advantage was demonstrated in OS. The study design was characterized by the short follow-up time (median 19.7 months) and allowed crossover to imatinib upon recurrence [2].

The second study, the Scandinavian Sarcoma Group (SSG) XVIII/German (AIO) trial, showed improvement of adjuvant imatinib for 3 years in terms of both RFS and OS in high-risk patients, when compared to 1-year treatment, with an HR of 0.55 (95% CI 0.37–0.83) and of 0.46 (95% CI 0.32–0.65) in OS and DFS, respectively [3].

Finally, the EORTC/Inter-group trial compared 2-year imatinib adjuvant treatment versus observation alone in radically resected GIST with an intermediate or high risk of recurrence. No statistically significant differences were observed between the two groups in terms of imatinib failure-free survival (IFFS), considered as an overall survival surrogate. Imatinib did not improve either IFFS (HR = 0.87, 95.7% CI 0.65–1.15) or OS (HR = 0.88, 95% CI 0.65–1.21) [20] (Table 3).

Mutational analysis of KIT and PDGFR-α should be assessed for the definition of adjuvant therapy to identify patients harboring less-susceptible mutations. In particular, patients with the KIT exon 11 deletion mutation benefit most from the longer duration of adjuvant imatinib, while patients with the kit exon 9 mutation could benefit more from adjuvant imatinib with 800 mg/day dose (based on activity data in patients with advanced GIST in this specific genotype). At present, however, no clinical prospective study has been conducted that supports this indication [28,29]. On the other hand, the PDGFR-α D842V mutation appears to be insensitive to imatinib, as well as wild-type, SDH, and NF1 mutant GIST [30]. In this light, the evaluation of specific molecular subtypes is the cornerstone for the clinical management of GIST from localized to metastatic disease, given their relevance to predicting clinical behavior and the response to molecularly targeted agents.

### 3.3. Future Directions

Despite, to date, 3 years of imatinib remaining the standard of care in the adjuvant setting, the main open question is defining the optimal duration of the adjuvant therapy. Two studies are currently enrolling to evaluate whether prolonging imatinib therapy to 5 or 6 years is safe and effective (NCT02413736 and NCT02260505). In addition, a new molecule was evaluated. A highly selective TKI against c-Kit and PDGFR, masitinib, has been tested in completely resected GIST with a high risk of recurrence (NCT02009423) but the study has been discontinued due to a sponsor decision (Table 4).

## 4. Non-Small Cell Lung Cancer (NSCLC)

### 4.1. Epidemiology and Prognosis

Almost one-third of patients are diagnosed with resectable NSCLC. The 5-year survival rates for NSCLC may vary from 73% in stage IA disease to 13% in stage IV disease. The most crucial factor for predicting recurrence rates and prognosis is stage followed by tumor histology/molecular profile, age, and performance status. Currently, adjuvant chemotherapy is indicated in the presence of nodal metastasis or primary tumor with a maximum diameter bigger than 4 cm. However, adjuvant chemotherapy marginally improves DFS and OS (absolute benefits of 5.8% and 5.4% after 5 years, respectively), with a high recurrence rate (up to 50% after 5 years) [31].

Targeted therapies are already an established reality in the treatment of advanced NSCLC [32,33,34,35,36,37,38,39]. Targetable alterations include mutations in KRAS (20–30%), EGFR (10–15% of Caucasian patients and up to 40% of Asian patients), BRAF (2–4%), HER2 (1–2%), and MET (2–4%), and rearrangements in ALK (3–7%), ROS1 (1–2%), RET (1–2%), and NTRK (0.5–1%).

To date, only patients with EGFR mutant NSCLC can benefit from adjuvant TT, while several studies are currently ongoing to define the role of adjuvant TT in other molecular subgroups, as well as in the perioperative setting.

The main clinical trials exploring TT in the adjuvant setting are reported in Table 5, while selected ongoing trials are reported in Table 6.

### 4.2. Major Advances in the Adjuvant Setting

In 2005, Tsuboi et al., in a phase III trial compared gefitinib (a first-generation EGFR TKI) to placebo in resected NSCLC [40]. Despite safety and feasibility being reported, no survival data were available due to an early trial closure due to a rising rate of interstitial lung disease (ILD) in patients treated with gefitinib [47].

Later, Goss et al., reported no benefit in terms of DFS (HR = 1.28, 95% CI 0.92–1.76, *p* = 0.14) or OS (HR = 1.24, 95% CI 0.90–1.71, *p* = 0.18) of 1-year adjuvant gefitinib in resected NSCLC (stage IB-IIIA) not selected for EGFR activating mutations [41].

In 2013, 6 months of consolidation treatment with gefitinib after platinum-based adjuvant chemotherapy demonstrated an improvement in DFS (HR = 0.37, 95% CI 0.16–0.85, *p* = 0.014), in the absence of OS benefit, in patients with resected stage IIIA NSCLC harboring EGFR mutations [48]. One phase II trial evaluating icotinib (a first-generation EGFR TKI), for 4–8 months, after platinum-based adjuvant chemotherapy, in resected NSCLC EGFR mutant stage IB-II-IIIA showed no significant increase in the proportion of disease-free patients (90.5% vs. 66.7%, *p* = 0.06) compared to observation alone [49]. Similarly, the RADIANT phase III trial, comparing 2-year erlotinib (a first-generation EGFR TKI) vs. placebo in an NSCLC adjuvant setting with EGFR-expressing tumors, but not necessarily EGFR-mutated tumors, demonstrated no benefit in terms of DFS (HR = 0.90, 95% CI 0.74–1.10, *p* = 0.32) or OS (HR = 1.13, 95% CI 0.88–1.44, *p* = 0.33) [43].

In 2018, the phase II EVAN trial reported a significant benefit in terms of DFS in favor of 2-year erlotinib compared to platinum-based chemotherapy in radically operated stage II-III NSCLC patients [50]. In the same year, 2-year gefitinib compared to platinum-based chemotherapy demonstrated, in 222 patients, improvement in RFS (HR = 0.60, CI 95% 0.42–0.87, *p* = 0.005) but not in OS outcomes (HR = 0.92, 95% CI 0.62–1.36, *p* = 0.67) [42,51].

Recently, the phase III IMPACT trial reported no significant benefit for adjuvant gefitinib when compared to platinum-based chemotherapy either in terms of RFS (HR = 0.92; 95% CI 0.67–1.28, *p* = 0.63) or in terms of OS (HR=1.03, 95% CI 0.65–1.65, *p* = 0.89) [45], while the EVIDENCE study comparing 2-year icotinib with platinum-based adjuvant chemotherapy demonstrated a significant benefit, for patients treated with icotinib, in terms of RFS (HR = 0.36, 95% CI 0.24–0.55, *p* < 0.0001) but not in OS (HR = 0.91, 95% CI 0.42–1.94, *p* = 0.80) [46].

The phase III ADAURA study randomized 682 radically operated NSCLC patients with stage IB-IIIA (VII edition of TNM staging) and classic EGFR activating mutations (exon 19 deletion or L588R exon 21 mutations) to receive either osimertinib 80 mg once daily (a third-generation EGFR TKI) or placebo for 3 years. The study achieved its primary endpoint by increasing RFS in patients with stage II-IIIA (HR = 0.23, 95% CI 0.18–0.30), with 70% of patients alive and disease-free at 48 months. The benefit was observed both in patients who had received prior adjuvant chemotherapy treatment (HR = 0.29, 95% CI 0.21–0.39) and in patients who had not received chemotherapy (HR = 0.36, 95% CI 0.24–0.55). Regarding the tolerability profile, grade 3 or 4 adverse events occurred in 23% of patients treated with osimertinib and 14% of patients treated with placebo [44]. The findings recently presented at ASCO 2023 suggest that the use of adjuvant osimertinib can also significantly improve OS, based on a median follow-up period of approximately 5 years. Specifically, among patients with stage II to IIIA, osimertinib was found to reduce the risk of death by 51% (HR = 0.49, 95% CI 0.33–0.73, *p* = 0.0004) [52].

Based on the positive results of the ADAURA study, osimertinib is currently indicated for adjuvant treatment after complete tumor resection in adult patients with stage IB-IIIA NSCLC whose tumors have EGFR exon 19 deletions or exon 21 L858R substitution mutations (Table 5).

### 4.3. Future Directions

#### 4.3.1. EGFR Mutant NSCLC

Several trials are currently ongoing to better define the role of EGFR TKIs in resected NSCLC.

The ADAURA2 trial evaluates 3-year osimertinib vs. placebo in resected stage IA2 and IA3 EGFR mutant NSCLC (NCT05120349). The phase III ALCHEMIST trial compares 2-year erlotinib versus observation in patients with resected NSCLC stage IB-IIIA and EGFR mutation (NCT02193282). The efficacy of 6-month or 12-month icotinib following chemotherapy compared to chemotherapy alone in patients with resected stage IIA-IIIA NSCLC harboring an EGFR mutation is currently under evaluation in the ICTAN trial (NCT01996098).

Furmomertinib, a novel third-generation EGFR TKI, is currently under evaluation in the phase III study FORWARD in EGFR mutant patients with stage from IIA to IIIA NSCLC after complete resection (NCT04853342).

Finally, the APEX trial randomizes patients with EGFR mutant stage II-IIIA NSCLC following complete tumor resection, to evaluate the efficacy and safety of almonertinib, a third-generation EGFR TKI targeting both EGFR-sensitizing and T790M resistance mutations, combined with or without chemotherapy (NCT04762459).

Based on the positive results obtained from the ADAURA trial, the NEOADAURA trial is a phase III, randomized, controlled, three-arm, multi-center study evaluating the benefit and safety of neoadjuvant osimertinib alone or in combination with chemotherapy, versus standard-of-care chemotherapy alone, in patients with resectable EGFR mutant NSCLC (NCT04351555) (Table 6).

#### 4.3.2. Other Oncogene-Addicted NSCLC

To date, no prospective data regarding the role of anti-ALK TKIs in the adjuvant/neoadjuvant setting have been published.

The ALCHEMIST screening trial is currently evaluating crizotinib (a first-generation ALK TKI) versus observation for up to 24 months after completion of standard-of-care chemotherapy and/or radiotherapy in ALK-positive patients (NCT02194738). In the ALINA trial adjuvant alectinib (a second-generation ALK TKI) for 24 months is compared with adjuvant platinum-based chemotherapy after surgical resection in patients with ALK-positive stage IB-IIIA NSCLC (NCT03456076) [53].

The activity of perioperative treatment with alectinib is being evaluated in the phase II ALNEO trial. Oral alectinib for 56 days before and 96 weeks after the surgery will be administered in potentially resectable stage II-III ALK-positive NSCLC (NCT05015010) (Table 6).

LIBRETTO-432 is a phase III study, evaluating 3 years of selpercatinib (a RET TKI) versus placebo after definitive locoregional treatment (surgery or radiotherapy) in participants with stage IB-IIIA RET fusion-positive NSCLC. Event-free survival is the primary endpoint and the estimated primary completion date is 2028 (NCT04819100) (Table 6).

NAUTIKA1 is a phase II study evaluating the efficacy and safety of 2 years of several targeted therapies in patients with resectable NSCLC in stages from IB to IIIB. In particular, the authors will evaluate neoadjuvant/adjuvant targeted therapies in patients harboring ALK, ROS1, NTRK, BRAF, and RET activating mutations with alectinib, entrectinib, vemurafenib, cobimetinib, and pralsetinib therapy, respectively. To date, only preliminary safety data of neoadjuvant alectinib for ALK+ NSCLC have been presented [54]. A perioperative approach with 8-week pre-operative capmatinib followed by 3-year adjuvant capmatinib is being evaluated in the phase II Geometry-N trial, enrolling patients with MET exon 14 mutations and/or high MET amplification stage I-IIIA NSCLC (NCT04926831).

## 5. Breast Cancer

### 5.1. Epidemiology and Prognosis

Breast cancer (BC) represents the first diagnosis of cancer in women and the second-leading cause of cancer death; in the last five years, the incidence rates increased by 0.5 annually, mainly due to early-stage and hormone receptor (HoR) positive disease.

The 5-years relative survival rate for localized, regional, and distant disease in American females is 99%, 86%, and 29%, respectively [6]. Thus, an unmet medical need is to prevent the development of a metastatic, and incurable, disease. Besides the stage at diagnosis, the tumor immunophenotype represents a further relevant factor associated with patient prognosis. Indeed, the survival rates range from 94% for HoR-positive to 85% and 77% for human epidermal growth factor receptor 2 (HER2) positive and triple-negative (TN) disease, respectively. Notably, the advances in personalized treatment for early/locally advanced disease recently introduced significantly improved (and will more deeply impact on) outcomes in featured tumor subtypes.

The main clinical trials exploring targeted therapies in the adjuvant setting are reported in Table 7; Table 8, while selected ongoing trials are reported in Table 9.

### 5.2. Major Advances in the Adjuvant Setting

The therapeutic options for early breast cancer (eBC) have been improved in the last few years and the greater use of neoadjuvant therapy (NAT), in particular for HER2-positive and TN BC [77,78], led to consider residual disease as a risk factor, in addition to classic prognostic elements such as tumor histology, grade, stage, hormone receptors, and HER2 expression, in the choice of the best adjuvant treatment [79,80]. Indeed, the latest evidence in BC adjuvant therapies concerns escalation targeted strategies addressed to high-risk patients, defined by lymph node-positive disease, residual disease after neoadjuvant therapy, higher Ki67, and/or selected using a composite score, such as the clinical and pathological stage (CPS) and estrogen-receptor status and histologic grade (EG) (CPS + EG scoring system), and the analysis of pathogenic mutated genes [81].

Despite the survival improvements for HER2-positive disease with the introduction of anti-HER2 therapy, the rate of disease relapse after adjuvant trastuzumab is still estimated at 15 to 30%; therefore, the introduction of escalation targeted treatments, such as other monoclonal antibodies (mAbs), tyrosine kinase inhibitors, and antibody–drug conjugates (ADCs), was investigated to reduce the risk of relapse. Regarding HoR-positive HER2-negative disease, the cornerstone adjuvant treatment is represented by endocrine therapy (ET); recently, it was tested in addition to three targeted strategies against cyclin-dependent kinase 4 and 6, poly-adenosine-diphosphate–ribose-polymerase (PARP) and mTOR pathway, respectively. Conversely, triple-negative eBC is still lacking in adjuvant targeted therapies except for the BRCA-mutated subgroup.

#### 5.2.1. HER2-Positive BC

Trastuzumab, a monoclonal antibody targeting the extracellular domain of the HER2 protein, represents a milestone in eBC targeted adjuvant therapy, reducing after one year of treatment the relative risk of recurrence by 40% and the risk of death by 30% in HER2-positive disease [55,56,80,82,83]. Phase III trials exploring adjuvant HER2-targeted agents are summarized in Table 7, Table 8 and Table 9. Although in the FinHER trial 9 weeks of trastuzumab plus chemotherapy versus chemotherapy alone showed a benefit [58,59], different trastuzumab durations (2 years, 9 weeks, or 6 months of trastuzumab) compared to 1 year of trastuzumab did not show superiority or non-inferiority efficacy [60,61,62,82,84], except for the PERSEPHONE trial (HR = 1.07, 90% CI 0.93–1.24, *p* = 0.011) [63]. Thus, one year is the standard duration of adjuvant trastuzumab for HER2-positive BC.

Concerning other therapeutic anti-HER2 mechanisms, lapatinib, an inhibitor of HER1 and HER2, failed to improve DFS both in place of trastuzumab for 1 year in the TEACH trial [64] and in a concurrent or subsequent addiction to trastuzumab in the ALTTO trial [65].

In July 2017, neratinib, a TKI against ErbB and HER2/HER4, was approved by the FDA for extended adjuvant treatment of HER2-positive eBC, based on the results of the ExteNET phase III randomized trial [66]. The 1-year administration of neratinib after trastuzumab-based therapy for stage II-IIIC disease showed a statistically significant 5-year invasive disease-free survival (iDFS) compared to placebo (HR = 0.73, 95% CI 0.57–0.92, *p* = 0.0083); the benefit was greater for HoR-positive disease (probably due to the crosstalk between estrogen and HER2 receptor signaling), for patients that started the treatment within 1 year of the end of trastuzumab, and for patients that had not achieved pCR after neoadjuvant therapy. Neratinib also improved the central nervous system (CNS) relapse of disease and demonstrated a numerical improvement in OS with a gain of 2.1% at 8 years (HR = 0.79, 95% CI, 0.55–1.13, *p* = 0.203) for the HoR-positive population who started treatment within one year of stopping trastuzumab [85]. In the ExteNET trial, 39% of patients reported G3-G4 diarrhea so a preemptive antidiarrheal strategy is recommended in patients who are candidates for neratinib [86]. In clinical practice, the role of neratinib remains not clarified given that new standards of care were subsequently approved.

In 2017 the therapeutic adjuvant landscape of HER2-positive eBC was enriched by the approval of pertuzumab, a monoclonal antibody against HER2 that prevents HER2/HER3 dimerization inhibiting cell survival and proliferation signaling, in combination with trastuzumab for 1 year and chemotherapy for high-risk patients (node-positive or tumor diameter greater than 1 cm), based on the primary analysis of the APHINITY trial [67]. The 6-year and 8.4-year follow-up analysis showed a statistical iDFS benefit (HR = 0.72, 95% CI 0.59–0.87) only for node-positive patients independently from HoR status, while OS superiority was not reached [87,88]. In conclusion, the addition of pertuzumab to adjuvant chemotherapy plus trastuzumab could be reserved only for node positive high-risk patients [80]. The use of pertuzumab after a pCR achieved with dual blockage remains an open question. The TRHYPAENA and BERENICE trials suggested continuing pertuzumab plus trastuzumab to complete 1 year of treatment in patients that reached pCR, even if trastuzumab alone remains the standard of care [89,90].

Trastuzumab emtansine (T-DM1), an antibody–drug conjugate (ADC) where the cytotoxic agent, emtansine, functions as a microtubule inhibitor and is covalently linked to trastuzumab, was the first ADC approved in 2019 as adjuvant treatment for HER2-positive residual breast or axilla invasive disease after taxane and trastuzumab-based neoadjuvant therapy. The phase III KATHERINE trial showed an improvement in 3-year iDFS (HR = 0.50, 95% CI 0.39–0.64, *p* < 0.001) with a relative 50% reduction in risk of relapse after 14 cycles of T-DM1 compared with trastuzumab [68]. The benefit was consistent in all subgroups, but no difference was shown in CNS relapse [91]. De-escalation chemotherapy strategy with the use of T-DM1 plus pertuzumab against taxane with pertuzumab and trastuzumab, after anthracycline-based chemotherapy, failed to meet the iDFS primary endpoint in the phase III KAITLIN study for HER2-positive eBC (node-positive or node-negative, HoR-negative and tumor size >2.0 cm) [69]. T-DM1 is now considered the new standard in the adjuvant context for residual invasive disease (Table 7).

#### 5.2.2. Luminal (HER2-Negative) BC

Abemaciclib represents the first cyclin-dependent kinase 4 and 6 inhibitor (CDK4/6i) approved in combination with endocrine therapy (tamoxifen or an aromatase inhibitor) by the FDA in October 2021 as targeted adjuvant therapy for HoR-positive, node-positive eBC at high risk of recurrence and with a Ki67 ≥ 20% [92].

In the monarchE trial patients with four or more positive nodes, or one to three nodes and either a tumor size ≥ 5 cm, histologic grade 3, or central Ki-67 ≥ 20% who had completed the local and systemic treatment, were randomized to standard ET with or without abemaciclib (150 mg twice daily for 2 years). The iDFS primary endpoint was met at the first (HR = 0.75, 95% CI 0.60–0.93, *p* = 0.01) and second pre-planned analyses (HR = 0.664, 95% CI 0.578–0.762, *p* < 0.0001) in the intention-to-treat (ITT) population and pre-specified subgroups, and about 50% of patients reported grade ≥ 3 adverse events (AEs) [70,71].

International guidelines recommend abemaciclib for high-risk selected patients irrespective of the Ki-67 cut-off, even if the OS data are still immature [93].

The advantaged position of abemaciclib as a unique CDK4/6i approved in this scenario will probably soon be removed by ribociclib’s introduction in the early strategy treatment of BC.

Recently, the second interim analysis of the NATALEE trial was presented at the 2023 ASCO Annual Meeting, showing promising result for stage II-III BC adjuvant therapy [72].

Ribociclib for 3 years (400 mg, 3 weeks on/1 week off) plus ET was tested against ET alone in men and women with stage II-III BC at risk of recurrence. It is interesting to notice that node-negative patients were also included in the study; moreover, in the case of stage IIA (node negative), more additional high-risk inclusion criteria were required: grade 3 or grade 2 plus others including Ki67 ≥ 20% and, for the first time, genomic risk profiling and Oncotype DX recurrence score. The choice of 400 mg instead of 600 mg standard dose was justified by the authors for major tolerability, with an extended duration of 3 years for a prolonged cell cycle arrest. The addition of ribociclib to ET showed a statically significant absolute iDFS (primary endpoint) benefit of 3.3%, consistent between subgroups. However, at a median follow-up of 34 months, only 20.2% of participants in the experimental arm had completed 3 years of treatment, 56.8% had completed 2 years, and 74.7% of participants remained on the study treatment at data cutoff. We need a longer follow-up to confirm these results, even though they asked the important question of recognizing patients diagnosed with early stage BC that are really in need of an escalation treatment [72].

Likewise, palbociclib, another CDK4/6i, was tested in combination with ET in the adjuvant (2 years of treatment) and post-neoadjuvant (1 year) setting in the PALLAS [73] and PENELOPE-B [74] trials, respectively. PALLAS enrolled stage II-III HoR-positive HER2-negative patients with the same design as the monarchE, while the PENELOPE-B trial enrolled high-risk patients with residual disease (CPS + EG ≥ 3 score or ypN positive) after receiving taxane-based neoadjuvant therapy. Neither of them established the superiority of palbociclib over ET alone in terms of iDFS. Some hypotheses were generated to justify the difference between the two CDK4/6is tested, such as the higher rate of discontinuation and intermittent schedule of administration of palbociclib [94]; the idea of a lower-risk population enrolled in the PALLAS trial was rejected considering the same outcomes for the different subgroups [95].

The Olympia trial led to the PARP inhibitor olaparib indication for HoR-positive germinal BRCA1/2-mutated high-risk selected patients [76].

In this trial, a total of 325 HoR-positive patients (about 18% of each arm of the study population) were included after local and systemic standard treatment, selected based on residual invasive disease with a CPS + EG ≥ 3 or at least four nodes positive for surgery upfront, achieving an iDFS (HR = 0.68, 95% CI 0.40–1.13) and OS (HR = 0.897, 95% CI 0.449–1.784) improvement [96].

The abemaciclib and olaparib approvals opened an important question about the best adjuvant treatment choice for patients who were candidates for both drugs. The OS benefit and the power of targeting a specific gene alteration could justify the olaparib preference over abemaciclib, despite the absence of comparison data.

Finally, regarding mTOR inhibitors, the SWOG S1207 phase III trial results were recently presented [75]. Patients in four high-risk groups (including Oncotype DX recurrence score > 25 or MammaPrint high-risk category) after (neo)adjuvant chemotherapy were randomized to receive ET with or without everolimus: the primary iDFS (HR = 0.94, 95%CI 0.77–1.14, *p* = 0.52) and the secondary OS (HR = 0.97, 95 CI 0.75–1.26, *p* = 0.84) endpoints were not met (Table 8).

#### 5.2.3. Triple-Negative BC

Triple-negative eBC, despite the approval of chemo-immunotherapy for neoadjuvant therapy [97], remains an orphan of new targeted therapies in the adjuvant setting except for the PARP inhibitor olaparib for BRCA-mutated disease [76]. The Olympia trial showed that the addition of 1-year olaparib after surgery and (neo)adjuvant chemotherapy for germinal BRCA1/2-mutated high-risk HER2-negative selected patients improved iDFS (HR = 0.58, 99.5% CI, 0.41–0.82, *p* < 0.001) and OS (HR = 0.68, 98.5% CI 0.47–0.97, *p* = 0.009) leading to its approval by the FDA in March 2022 [76,96]. A residual invasive disease after neoadjuvant therapy and a node-positive or tumor diameter ≥ 2 cm after initial surgery were selected as inclusion criteria for TNBC; the specific subgroup analysis confirmed a statistically significant benefit for TN disease (iDFS: HR = 0.62, 95% CI 0.487–0.787; OS: HR = 0.64, 95% CI 0.459–0.884). The characteristics of phase III trials with targeted therapy for HoR and triple-negative BC disease are summarized in Table 7, Table 8 and Table 9.

For patients that should continue pembrolizumab after neoadjuvant therapy, there are currently no data to help clinicians in the choice of the best adjuvant treatment after residual invasive disease. Future combination trials are needed and an adjuvant combination of olaparib plus pembrolizumab could be considered an option for the acknowledgment of good safety [98] (Table 8).

### 5.3. Future Perspectives

Over the last five years, many new drugs and new strategies focused on high-risk disease have been developed for eBC adjuvant treatment. The iDFS is commonly accepted as a surrogate endpoint for early-stage disease but the OS remains an important endpoint to confirm the first approval of new drugs. Future strategies are focusing on a combination of different drugs, the introduction of new intelligent therapies, and, above all, on recognizing high-risk patients by applying new technologies based on precision medicine. Ongoing phase III studies with adjuvant targeted therapy for BC are summarized in Table 7, Table 8 and Table 9.

Targeted therapies are being tested in HER2-positive patients not achieving pCR post-neoadjuvant therapy such as trastuzumab–deruxtecan (T-DXd), an ADC of trastuzumab linked to deruxtecan, a topoisomerase I inhibitor payload, against T-DM1 in the DESTINY-Breast05 trial (NCT04622319). The CompassHER2 RD trial (NCT04457596) is evaluating the combination of T-DM1 plus tucatinib, a HER2-specific TKI, for residual invasive disease to reduce CNS relapse considering the intracranial response rate demonstrated in the metastatic setting [99], while in the ASTEFANIA trial (NCT04873362) T-DM1 is evaluated in combination with atezolizumab, a mAb anti-PD-L1, in a potential combined effect in the cancer–immunity cycle [100].

The phase III SASCIA (NCT04595565) and ASCENT-5 (NCT05633654) ongoing clinical trials are testing sacituzumab govitecan, an ADC composed of a mAb against TROP-2 and SN38-topoisomerase I inhibitor, against physician’s choice treatment in patients with residual invasive disease post-neoadjuvant therapy in HER2-negative BC, as a single agent, and in TNBC, in combination with pembrolizumab.

Regarding HoR-positive disease, an ongoing clinical trial is still assessing the CDK4/6i potential benefit: the eMonarcHER trial (NCT04752332) is evaluating abemaciclib plus ET in patients with triple-positive BC who had completed adjuvant HER2-targeted therapy. Other studies are including the genomic assay platform Oncotype DX to select high-risk patients that could benefit from a CDK4/6i in addition to ET both as replacement of standard chemotherapy for ribociclib in the ADAPTcycle (NCT04055493) as (neo)adjuvant therapy and as escalation treatment after standard adjuvant chemotherapy for abemaciclib in the ADAPTlate trial (NCT04565054).

The COGNITION-GUIDE (NCT05332561) seven-arm umbrella phase II ongoing trial is a model of a precision medicine approach in a curative setting: patients with residual disease after neoadjuvant therapy and eventually after standard adjuvant therapy are allocated to a genomics-guided therapy determined by molecular alterations. Finally, futuristic approaches are evaluating the so-called “second-line” adjuvant treatments using circulating tumor DNA detection as a follow-up strategy to detect minimal residual disease (MRD) [101] and avoid a macroscopical relapse of disease: the DARE (NCT04567420) and the LEADER (NCT03285412) phase II trials are testing palbociclib and ribociclib in combination with ET for MRD in HoR-positive BC, respectively (Table 9).

## 6. Discussion

Targeted therapy is currently a consolidated reality in the treatment of advanced oncogene-addicted cancers, demonstrating significant improvement in survival and disease control outcomes. On the other hand, targeted therapy in the adjuvant setting is assuming an ever-increasing role and represents, to date, a treatment option in selected patients with potentially cured disease [3,4,45]. However, further research is still needed, and some questions remain open.

DFS and OS have been considered as primary and secondary endpoints, respectively, in almost all trials that led to the approval of TT in the adjuvant setting [3,4,45,95] (Figure 1).

OS has traditionally been considered the most clinically meaningful endpoint in clinical trials because it provides a comprehensive measure of the treatment overall effectiveness. However, in the context of early-stage cancer treatment, where perioperative or adjuvant therapies are evaluated, OS may be confounded by the effects of multiline therapy after tumor recurrence and may not fully capture the benefits of the treatments.

Therefore, alternative endpoints, such as DFS and other surrogate endpoints, are considered.

A meta-analysis conducted by Suciu et al., in 2018 aimed to evaluate whether DFS could be considered a valid surrogate endpoint for OS in resected stage II–III melanoma patients who were receiving adjuvant therapy with interferon or checkpoint inhibitors [102]. The study found that DFS was a valid surrogate endpoint for OS in this population and that improvements in DFS were associated with improvements in OS [102].

Another meta-analysis also evaluated the correlation between DFS and OS as endpoints in patients with resectable locally advanced NSCLC. The authors, analyzing a large sample of data for patients with resected NSCLC cancer, suggest DFS as a valid surrogate endpoint for OS in studies of adjuvant chemotherapy [103]. Notably, no data about adjuvant TT are collected in this study [103]. In addition, evaluation of OS data in early-stage cancer clinical trials can be challenging due to the lengthy follow-up duration required to observe enough events. These patients demonstrated a better prognosis than those with advanced disease, and, consequently, it may take significantly more time to gather an adequate number of occurrences for OS analysis.

Furthermore, no data about treatments used at the time of relapse have been reported (e.g., in ADAURA [44] and NCT00116935 [3]), thus leaving, at least, two open questions. *How is OS influenced by the therapies at progression? What treatment can clinicians use at the time of disease relapse?* While the first question remains still unanswered and more studies will be necessary, the second may find an answer according to cancer histology.

Currently, in some histologies, e.g., breast cancer, the duration after completing additional therapy can be taken into account when considering the possibility of reintroducing treatment or altering the treatment approach [104]. However, in other cancers, such as NSCLC, determining the most suitable therapy upon relapse can be more tangled. In this light, managing a patient with EGFR mutant NSCLC experiencing a relapse shortly after completing adjuvant osimertinib therapy can pose challenges, due to the insufficient evidence demonstrating the superiority of initiating first-line polychemotherapy instead of rechallenging. Borrowing the evidence from the advanced setting where molecular mechanisms of resistance to EGFR TKIs may include various genetic alterations (such as MET amplification, HER2 amplification, PIK3CA alterations, BRAF mutation, and KRAS mutation [105]), tissue or/and liquid rebiopsy could be useful to identify any molecular mechanisms responsible for resistance to TT. In the context of melanoma, evidence from studies on metastatic disease demonstrates that resistance mechanisms may manifest within the PI3K or NRAS signaling pathways. These data may potentially guide a future approach with targeted therapies (such as PI3K inhibitors) within the adjuvant setting or for effectively addressing recurrence [106]. On the other hand, concerning HER2-positive breast cancer, the emergence of recurrence during or following adjuvant therapy may possibly be attributed to a spectrum of molecular alterations, including the activation of the PI3K/AKT pathway, PTEN loss, or heightened ER activation. Each of these alterations could presents an intriguing opportunity for therapeutic intervention to be explored in dedicated clinical trials. PI3K inhibitors, mTOR inhibitors, and various endocrine therapies could emerge as valuable tools that can be harnessed to specific therapeutic targets [107].

TT is generally well-tolerated with a significantly better safety profile compared, in particular, to chemotherapy [108]. Nevertheless, these types of treatments are not exempt from potential side effects. Grade 3–4 adverse events are reported in percentages ranging from 23% to 41%, by studies evaluating TT in the adjuvant setting [4,45]. The exposure of potentially cured patients to treatments that can lead to non-negligible toxicity is justifiable to improve DFS and OS. However, in many cases (such as adjuvant imatinib or osimertinib) the duration of such therapies is not yet defined with certainty. Two studies are currently evaluating (and currently enrolling) whether prolonging imatinib therapy to 5 or 6 years is safe and effective (NCT02413736 and NCT02260505). On the other hand, the duration of adjuvant osimertinib, in the ADAURA study [44], was arbitrarily set at 3 years and probably influenced by the ADJUVANT trial [43,50]. Considering these premises, it is crucial to gain a deeper understanding of which patients are most likely to derive significant benefits from adjuvant TT.

The use/implementation of liquid biopsy, which involves the analysis of circulating tumor cells (CTCs) or circulating tumor DNA (ctDNA) in the blood to provide real-time monitoring of disease recurrence and to potentially guide treatment decisions, represents a promising research area, not only in advanced but also in early stages.

Abbosh et al., developed a patient-specific mutational panel assay using next-generation sequencing (NGS) technology on plasma samples, consisting of 12–30 single-nucleotide variants (SNVs). ctDNA was longitudinally evaluated in 24 patients with resected NSCLC, demonstrating that the detection of SNVs was associated with clinical evidence of lung cancer relapse [109]. Moreover, Chen et al., used the Oncomine Research Panel (134 cancer-related genes assay) to evaluate the potential of ctDNA as a biomarker of MRD in patients with TNBC who had residual disease after neoadjuvant chemotherapy. Despite the low detection rate of ctDNA mutations, all four patients with detectable ctDNA experienced disease relapse within 9 months [110].

Through the analysis of CTCs and ctDNA, physicians can gain insights into drug sensitivity or resistance and residual tumor cells, both during and after adjuvant therapy [111].

Finally, in recent years, neoadjuvant TT has emerged as a promising approach in early-stage cancer treatment. While for some histologies neoadjuvant TT is already a well-established reality (e.g., breast cancer), for some others, several studies are still ongoing. One of the main advantages is that neoadjuvant therapy allows the assessment of tumor response to treatment before surgery. This provides important information on the sensitivity of the tumor to the therapy, which can be used to guide subsequent treatment decisions.

In addition, in some histologies (e.g., HER2-positive breast cancer), neoadjuvant TT plus chemotherapy has been associated with a higher rate of pCR and better survival outcomes compared to neoadjuvant chemotherapy alone [112].

Regarding lung cancer, several studies, e.g., NeoADAURA [113] and ALNEO [114], are currently evaluating TT in a perioperative approach in resectable NSCLC.

## 7. Conclusions

In conclusion, adjuvant TT has emerged as a promising treatment option, with the potential for significant improvements in survival and disease control outcomes. In addition to historically proven effective therapies such as trastuzumab in breast cancer and imatinib in GIST, novel therapeutic approaches are progressively emerging, increasingly demonstrating promising outcomes. Examples include adjuvant BRAF inhibitors in melanoma, osimertinib in resected NSCLC, and abemaciclib in breast cancer, with even more recent advancements seen in perioperative strategies across diverse histologies. Ongoing research on the development of new therapies and understanding novel biomarkers will be crucial in realizing this potential and further enhancing the possibilities of early-stage cancer treatment.

## Figures and Tables

**Figure 1 jpm-13-01427-f001:**
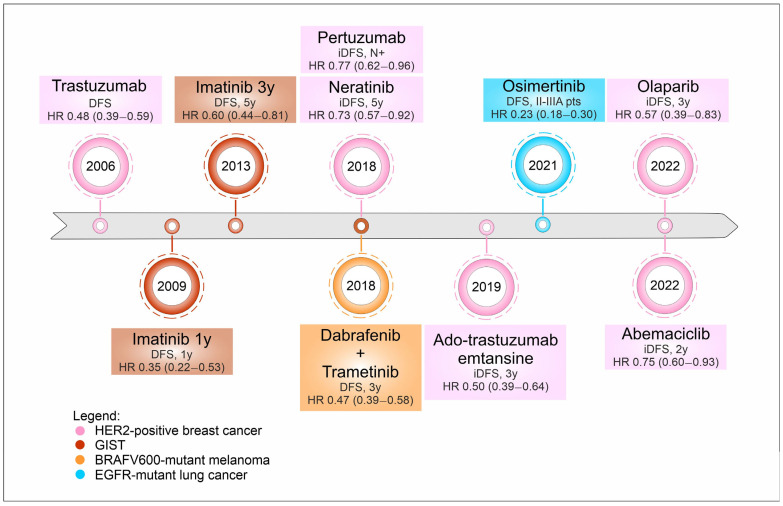
Timeline of targeted therapies’ approval in the last decade across several histologies. Legend: DFS, disease free survival; HR, hazard ratio; y, years; iDFS, invasive disease-free survival; N+, nodal involvement; pts, patients.

**Table 1 jpm-13-01427-t001:** Characteristics and results of the main clinical trials exploring targeted therapies in resected melanoma.

Trial	Phase	Setting	Stage	Study Arm(s)	Target	N	Primary Endpoint	Main Results	Safety (AEs Grade 3–4)
Long et al. [4] (Combi-AD) (NCT01682083)	III	Adjuvant	IIIA, IIIB, IIIC	Dabrafenib–trametinib vs. placebo for 1 year	BRAF + MEK	870	DFS	3 y DFS 58% (D+T) 39% (P) HR = 0.47 (0.39–0.58) *p* < 0.001 3 y OS 86% (D+T) 77% (P) HR = 0.57 (0.42–0.79) *p* = 0.0006	41% (D+T) 14% (P)
Maio et al. [9] (BRIM-8) (NCT01667419)	III	Adjuvant	Cohort 1: IIC, IIIA, IIIB Cohort 2: IIIC	Vemurafenib vs. placebo for 1 year	BRAF	Cohort 1: 364 Cohort 2: 184	DFS	Cohort 1: mDFS NR (V) 36.9 mo (21.4-NE) HR = 0.54 (0.37–0.78) *p* = 0·0010 Cohort 2: mDFS 23.1 mo (18.6–26.5) (V) 15.4 mo (11.1–35.9) (P) HR = 0.80 (0.54–1.18) *p* = 0.026	57% (V) 15% (P)
Amaria et al. [10] (COMBI-Neo) (NCT02231775)	II	Neoadjuvant/Adjuvant	III, oligometastatic IV	Dabrafenib–trametinib (8 wks neoadjuvant + 44 wks adjuvant) vs. SOC	BRAF + MEK	21	EFS	mEFS 19.7 mo (16.2-NE) (D + T) 2.9 mo (1.7-NE) (SoC) HR = 0.016 (0.00012–0.14) *p* < 0.0001	G3 diarrhea 15% Other 6 G3 AEs (each 8%)
Long et al. [11] (Neo-COMBI) (NCT01972347)	II	Neoadjuvant/Adjuvant	IIIB-C	Dabrafenib–trametinib (12 wks neoadjuvant + 40 wks adjuvant)	BRAF + MEK	35	pCR	pCR 49% (31–66) non-pCR 51% (34–69)	29%

Legend: SOC, standard of care; Wks, weeks; DFS, disease-free survival; EFS, event-free survival; pCR, pathological complete response; HR, hazard ratio; y, years; D+T, dabrafenib + trametinib; OS, overall survival; P, placebo; V, vemurafenib; AE, adverse event; mo, months; NE, not estimable; NR, not reached.

**Table 2 jpm-13-01427-t002:** Ongoing clinical trial exploring targeted therapies in resected melanoma.

Clinical Trial Identifier (Trial Name)	Design	Number of Patients (N)	Stage	Study Arms	Target	Duration TKIs	Primary Endpoint	Status
NCT05270044 (Columbus-AD)	Phase III, randomized, triple-blinded, 2-arm	815	IIA/B/C	Encorafenib + binimetinib vs. placebo	BRAF + MEK	1y	DFS	Recruiting

Legend: DFS, disease-free survival; Y, year.

**Table 3 jpm-13-01427-t003:** Characteristics and results of the main clinical trials exploring targeted therapies in resected GIST.

Trial	Phase	Stage	Study Arm(s)	Target	N	Primary Endpoint	Main Results	Safety (AEs Grade 3–4)
Dematteo et al. [2] (Z9001 trial) (NCT00041197)	III	>3 cm	Imatinib vs. placebo for 1 year	c-Kit	713	DFS	1 y DFS 98% (I) 83% (P) HR = 0.35 (0.22–0.53) *p* < 0.0001 1 y OS 99.2% (I) 99.7% (P) HR = 0.66 (0.22–2.03) *p* = 0.4714	18.3 (P) 30.9% (I)
Joensuu et al. [3] (NCT00116935)	III	High Risk	Imatinib—1 vs. 3 years	c-Kit	400	DFS	5 y DFS: 71.1% (3y) 52.3% (1y) HR = 0.60 (0.44–0.81) *p* = 0.001 5 y OS: 91.9% (3y) 85.3% (1y) HR= 0.60 (0.37–0.97) *p* = 0.036	32.8% (3y) 20.1% (1y)
Casali et al. [20] (NCT00103168)	III	High/Intermediate Risk	Imatinib vs. placebo for 2 years	c-Kit	835	IFFS	10 y IFFS 75% (I) 74% (P) HR = 0.87 (0.65–1.15) *p* = 0.31 10 y DFS 63% (I) 61% (P) HR = 0.71 (0.57–0.89) *p* = 0.002 10 y OS 80% (I) 78% (P) HR = 0.88 (0.65–1.21) *p* = 0.43	15.4% (I)

Legend: DFS, disease-free survival; IFFS, imatinib failure-free survival; HR, hazard ratio; P, placebo; I, imatinib; Y, years; OS, overall survival.

**Table 4 jpm-13-01427-t004:** Ongoing clinical trials exploring targeted therapies in resected GIST.

Clinical Trial Identifier	Design	Number of Patients (N)	Stage	Study Arms	Target	Duration TKIs	Primary Endpoint	Status
NCT02413736	Phase III, randomized, open label, 2-arm	250	High Risk	Imatinib	c-Kit	3 vs. 5 y	DFS	Recruiting
NCT02260505	Phase III, randomized, open label, 2-arm	134	High Risk	Imatinib	c-kit	3 vs. 6 y	DFS	Recruiting
NCT02009423	Phase III, randomized, double-blinded, 2-arm	7	High Risk	Masitinib vs. Placebo	c-kit PDGFR	2 y	DFS	Terminated by sponsor

Legend: DFS, disease-free survival.

**Table 5 jpm-13-01427-t005:** Characteristics and results of the main clinical trials exploring targeted therapies in resected non-small cell lung cancer.

Trial	Phase	Stage	Study Arm(s)	Target	N	Primary Endpoint	Main Results	Safety (AEs Grade 3–4)
Tsuboi et al. [40] (NCT02511106)	III	IB−IIIA	Gefitinib vs. placebo (2y)	EGFR	38 (UP)	NA	NA	NA
Goss et al. [41] (BR19) (NCT00049543)	III	IB−IIIA	Gefitinib vs. placebo (2y)	EGFR	503 (UP) 15 (EGFRm)	OS	mOS 5.1 y (4.4-NE) (G)—NR (P) HR = 1.24 (0.94–1.64), *p* = 0.14 mDFS 4.2 y (3.2-NE) (G)—NR (P) HR = 1.22 (0.93–1.61), *p* = 0.15	5–8% (G) (mainly rash, diarrhea, dyspnea)
Zhong et al. [42] (CTONG1104/ ADJUVANT) (NCT01405079)	III	II−IIIA (N1-2)	Gefitinib (2 y) vs. ChT (4 cycles)	EGFR	222 (EGFR m)	DFS	DFS 28.7 mo (G)—18 mo (ChT) HR = 0.60 (0.42–0.87), *p* = 0.0054 mOS 75.5 mo (G)—79.2 mo (ChT) HR = 0.96 (0.64–1.43), *p* = 0.823	12% (G) 48% (ChT)
Kelly et al. [43] (RADIANT) (NCT00373425)	III	IB−IIA	Erlotinib vs. placebo (2y)	EGFR	973 (EGFR exp) (161 EGFRm)	DFS	mDFS 50.5 mo (E)—48.2 mo (P) HR = 0.90 (0.74–1.10) *p* = 0.324 mOS NR (E)—NR (P) HR = 1.13 (0.881–1.448) *p* = 0.335 In EGFRm pts mDFS 46.4 (E)—28.5 mo (P) HR = 0.61 (0.384–0.981)	22.3% (rash) (E) 6.2% (diarrhea) (E)
Herbst et al. [44] (ADAURA) (NCT02511106)	III	IB−IIIA	Osimertinib (3 y) vs. placebo (3 y)	EGFR	682 (EGFRm)	DFS in II-IIIA pts	DFS (II-IIIA) 65.8 (O)—21.9 mo (P) HR = 0.23 (0.18–0.30) DFS (OP) 65.8 (O)—28.1 mo (P) HR = 0.27 (0.21–0.34)	23% (O) 14% (P)
Tada et al. [45] (IMPACT) (UMIN000006252)	III	II-IIIA	Gefitinib (2 y) ChT (4 cycles)	EGFR	232 (EGFR m)	DFS	DFS 35.9 mo (G)—25.0 mo (ChT) HR = 0.92 (0.67–1.28), *p* = 0.63 5 y survival rates: 78.0% (G) vs. 74.6% (ChT) HR = 1.03 (0.65–1.65), *p* = 0.89	NA
He et al. [46] (EVIDENCE) (NCT02448797)	III	II-IIIA	Icotinib (2y) vs. ChT (4 cycles)	EGFR	322 (EGFR m)	DFS	mDFS 47.0 mo (I)—22.1 mo (ChT) HR = 0.36 (0.24–0.55), *p* < 0.0001 3y-DFS 63.9% (51.8–73.7) (I)—32.5% (21.3–44.2) (ChT)	-Rash (2%) (I) -Neutropenia (41%) (ChT) -Leukopenia (19%) (ChT) -Vomiting (13%) (ChT) -Nausea (7%) (ChT)

Legend: ChT, chemotherapy; UP, unselected patients; EGFRexp, EGFR expressing; EGFRm, EGFR mutant; NA, not available; DFS, disease-free survival; OS, overall survival; HR, hazard ratio; RR, risk ratio; pts, patients; OP, overall population; y, years; E, erlotinib; O, osimertinib; I, icotinib; P, placebo.

**Table 6 jpm-13-01427-t006:** Ongoing clinical trials exploring targeted therapies in resected non-small cell lung cancer.

Clinical Trial Identifier	Design	Setting	Driver Mutation	Estimated Number of Patients (N)	Stage	Study Arms	Duration TKIs	Primary Endpoint	Status
NCT01996098 (ICTAN)	Phase III, randomized, open label, 3-arm	Adjuvant	EGFR activating mutation in exon 19 or 21	318	IIA-IIIA	Icotinib + chemo (6 mo) vs. icotinib + chemo (12 mo) vs. chemo alone	6 mo vs. 12 mo	DFS	Unknown
NCT05120349 (ADAURA2)	Phase III, randomized, triple blind, 2-arm	Adjuvant	EGFR Ex19Del L858R	380	IA2, IA3	Osimertinib vs. placebo	3 y	DFS	Recruiting
NCT04853342 (FORWARD)	Phase III, randomized, double blind, 2-arm	Adjuvant	EGFR Ex19Del L858R	318	II-IIIA	Furmonertinib versus placebo	NA	DFS	Not yet recruiting
NCT04762459 (APEX)	Phase III, randomized, open label, 3-arm	Adjuvant	EGFR Ex19Del L858R	606	II-IIIA	Almonertinib vs. almonertinib + chemo vs. Chemo alone	3 y	DFS	Enrolling by invitation
NCT02193282 (ALCHEMIST)	Phase III, randomized, 4-arm	Adjuvant	EGFR Ex19Del L858R	450	IB (≥4cm)-IIIA	Erlotinib vs. placebo (blinded) vs. erlotinib vs. placebo (unblinded)	2 y	OS	Active, not recruiting
NCT03381066	Phase III, randomized, open label, 2-arm	Adjuvant	EGFR Ex19Del L858R	225	IIa-IIIb (excluding N3)	Chemo + gefitinib vs. chemo	1 y	DFS	Unknown
NCT04687241	Phase III, randomized, triple blind, 2-arm	Adjuvant	EGFR Ex19Del L858R	192	II-IIIB	Almonertinib vs. placebo	NA	DFS	Active, not recruiting
NCT02125240	Phase III, randomized, quadruple blind, 2-arm	Adjuvant	Sensitive EGFR gene mutation (19/21)	124	II-IIIA	Icotinib vs. placebo	NA	DFS	Unknown
NCT04351555 (NEOADAURA)	Phase III, randomized, double blind, 3-arm	(Neo) Adjuvant	EGFR Ex19Del L858R	328	II-IIIB N2	Placebo + chemo vs. osimertinib + chemo vs. osimertinib alone	≥9 weeks	MPR	Recruiting
NCT03456076	Phase III, randomized, open label, 2-arm	Adjuvant	ALK	257	IB-IIIA	Alectinib vs. chemo alone	2 y	DFS	Active, not recruiting
NCT05341583	Phase III, randomized, quadruple blind, 2-arm	Adjuvant	ALK	202	II-IIIB	Ensartinib vs. placebo	2 y	DFS	Recruiting
NCT04819100 (LIBRETTO-432)	Phase III, randomized, triple blind, 2-arm	Adjuvant	RET	170	IB-IIIA	Selpercatinib vs. placebo	3 y	EFS	Recruiting

Legend: Chemo, chemotherapy; DFS, disease-free survival; OS, overall survival; MPR, major pathological response; EFS, event-free survival; wks, weeks.

**Table 7 jpm-13-01427-t007:** Characteristics and results of the main clinical trials exploring targeted therapies in resected HER2-positive breast cancer.

Trial	Phase	Stage	Study Arm(s)	N	Primary Endpoint	Main Results	Safety (AEs Grade 3–4)
Piccart-Gebhart et al. [55] (HERA) (NCT00045032)	III	I–IIIC (node negative T ≥ 1 cm)	ChT (4 cycles) vs. ChT followed by H (1 year) vs. ChT followed by H (2 years)	5081	DFS	H 1 year vs. observation DFS: HR 0.54 (0.43–0.67), *p* < 0.0001 OS: HR: 0.66 (0.47–0.91) *p* = 0.015 11-year follow-up H 1 year vs. H 2 years HR 1.02 (0.89–1.17)	7.9% (H 1y) vs. 4% (ChT)
Romond et al. [56] (NSABP B-31/NCCTG N9831) (NCT00005970)	III	pN positive Only for NCCTG N9831: pN negative with at least one of the following: T ≥ 2 cm if HoR positive T ≥ 1 cm if HOR negative	AC-TXL +/− H (52 wks)	3351	DSF	DFS: HR 0.48 (0.39–0.59), *p* < 0.001 OS: HR 0.67 (0.48–0.93) *p* = 0.015	Class III or IV congestive heart failure: NSABP-B31 4.1% (H) vs. 0.8% (ChT) N9831 2.9% vs. 0%
Slamon et al. [57] (BCIRG 006) (NCT00021255)	III	Stage I–III	AC-TXT +/− H (52 wks) vs. TXT/carbo/H (52 wks)	3222	DFS	AC-TXT vs. AC-TXT+H 5-year-DFS: HR 0.64, *p* < 0.001 5-year-OS: HR 0.63, *p* < 0.001 AC-TXT vs. TXT/Carbo/H 5y-DFS: HR 0.75, *p* = 0.04 5y-OS: HR 0.77, *p* = 0.04	Class III or IV congestive heart failure: 2% (AC-TXT-H) 0.7% (AC-TXT) 0.4% (TXT/carbo/H
Joensuu et al. [58] (FinHer) (ISRCTN76560285)	III	HER2-positive subgroup: pN positive or pN negative with T ≥ 2 and PgR negative	TXT→FEC vs. TXT/H (9 wks)→FEC vs. V → FEC vs. V/H (9 wks)→FEC	1010 232 HER2-positive	DDFS	RFS: HR 0.42 (0.21–0.83) *p* = 0.01 OS: HR 0.41 (0.16–1.08) *p* = 0.07 5-year DDFS ChT+H vs. ChT HR 0.65 (0.38–1.12), *p* = 0.12 TXT/H -> FEC vs. TXT-> FEC HR 0.32 (0.12–0.89), *p* = 0.029	100% (TXT/H) 75.9% (V/H)
Joensuu et al. [59] (SOLD) (NCT00593697)	III	Stage I–III	TXT/H (9 wks)→ FEC vs. TXT/H (9 wks)→ FEC→ H (42 wks)	2176	DFS	HR 1.39 (1.12–1.72)	56% (H 9 wks) vs. 58% (H 1 year) Cardiac adverse event: 2% (H 9 wks) 4% (H 1 year)
Pivot et al. [60] (PHARE) (NCT00381901)	III	Stage I–III	After at least 4 cycles of ChT: H (6 months) vs. H (1 year)	3380	DFS	HR 1.28 (1.05–1.56), *p* = 0.29	Cardiac events (any grade) 5.7% (1 year) vs. 1.9% (6 months)
Conte et al. [61] (Short-HER) (NCT00629278)	III	Stage I–IIIC (if pN0 at least one of: pT > 2 cm, G3, lympho-vascular invasion, Ki-67 > 20%, age ≤35 years, or HoR negative	TXT/H (9 wks)→FEC vs. AC→TXL or TXT/H (1 year)	1254	DFS (non-inferiority)	HR 1.13 (0.89–1.42) (non-inferiority margin set at 1.29)	Cardiac events 1.3% (H 9 wks) vs. 2.9 (H 1 year)
Mavroudis et al. [62] (HORG) (NCT00615602)	III	pN positive or high-risk pN negative	ChT/H (6 months) vs. ChT/H (1 year)	481	DFS (non-inferiority)	DFS HR 1.57 (0.86–2.10); *p* = 0.137 (non-inferiority margin set at 1.53)	Cardiotoxicity 0.8% (H 6 months)
Earl et al. [63] (PERSEPHONE) (NCT00712140)	III	Stage I–III	ChT/H (6 months) vs. ChT/H (1 year)	4088	DFS	4y-DFS: HR 1·07 (0·92–1·24) *p* = 0.023 for non-inferiority, *p* = 0.49 for superiority 4y-OS: HR 1.13 (0.94–1.37); *p* = 0.017 for non-inferiority, *p* = 0.27 for superiority	24% (H 1 year) 19% (H 6 months)
Goss et al. [64] (TEACH) (NCT00374322)	III	Stage I–III	Lapatinib vs. placebo (1 year)	3161	DFS	HR 0.83 (0.70–1.00), *p* = 0.053	Diarrhea: 6% (L) vs. 1% (P)
Piccart-Gebhart et al. [65] (ALTTO) (NCT00490139)	III	Stage I–III (if N negative T > 1 cm)	Lapatinib vs. lapatinib/H vs. LH-lapatinib vs. H	8381	DFS	Lapatinib/H vs. H HR 0.84 (0.70–1.02), *p* = 0.48 H→lapatinib vs. H HR 0.96(0.80–1.15), *p* = 0.61	46% (L/H) 32% (H/L) 41% (L) 25% (H)
Martin et al. [66] (ExteNET) (NCT00878709)	III	Stage II–III	Neratinib vs. placebo (1 year)	2840	iDFS	5y-iDFS 90.2% (neratinib)–87.7 (placebo) HR 0.73 (0.57–0.92), *p* = 0.0083	Diarrhea:40% (neratinib) vs. 2% (P) Nausea and vomiting: 2–3% (neratinib) vs. <1% (P)
Von Minckwitz et al. [67] (APHINITY) (NCT01358877)	III	Stage I–III	ChT with Pert/H vs. ChT with H/placebo	4804	iDFS	ITT: HR 0.81 (0.66–1.00) *p* = 0.045 N positive population: HR 0.77 (0.62–0.96), *p* = 0.02 N negative population: HR 1.13 (0.68–1.86), *p* = 0.64	64.2%(Pert) vs. 57.35% (H)
Von Minckwitz et al. [68] (KATHERINE) (NCT01772472)	III	Residual disease after NACT (taxane + H ± anthracycline)	T-DM1 vs. H (14 cycles)	1486	iDFS	3 y iDFS: 88.3% (T-DM1) vs. 77.0% (H) HR 0.50; (0.39–0.64), *p* < 0.001	25.7% (Pert) vs. 15.4% (H)
Krop et al. [69] (KAITLIN) (NCT01966471)	III	Stage II (with N positive or HoR negative) III	After 3–4 cycles of anthracycline-based ChT: T-DM1/pert vs. taxane/pert/H	1846 (1658 node positive)	iDFS	iDFS N positive: HR 0.97; (0.71–1.32), *p* = 0.83 iDFS overall population: HR 0.98 (0.72–1.32)	55.4% (T-DM1/Pert) vs. 51.8% (Pert/H)

Legend: Ph, phase; AC, doxorubicin/cyclophosphamide; ChT, chemotherapy; DDFS, distant disease-free survival; FEC, fluorouracil, epirubicin, and cyclophosphamide; G, grading; H, trastuzumab; HoR, hormone receptor; iDFS, invasive disease-free survival; L, lapatinib; N, node; NACT, neoadjuvant chemotherapy; OS, overall survival; P, placebo; Pert, pertuzumab; pN, pathological node; PgR, progesterone receptor; T, tumor size; TXL, paclitaxel; TXT, docetaxel; V, vinorelbine; wks, weeks.

**Table 8 jpm-13-01427-t008:** Characteristics and results of the main clinical trials exploring targeted therapies in resected hormone-positive and triple-negative breast cancer.

Trial	Phase	Subtype	Stage	Study Arm(s)	Target	N	Primary Endpoint	Main Results	Safety (AEs Grade 3–4)
Stephen et al. [70,71] (monarchE) (NCT03155997)	III	HoR-positive	≥N2 or ≥N1 with at least one of the following: G3, T ≥ 5 cm, or Ki 67 > 20%	ET/abemaciclib vs. ET/placebo (2 years)	CDK4/6	5637	iDFS	2 y iDFS: 92.2% (A) vs. 88.7% (P) HR 0.75 (0.60–0.93), *p* = 0.01	45.2% (A) vs. 12.7% (P)
Slamon et al. [72] (NATALEE) (NCT03701334)	III	HoR-positive	Stage II–III	ET/ribociclib vs. ET/placebo (3 year)	CDK4/6	5101	iDFS	3 y iDFS: 90.4 (RB) vs. 87.1% (P) HR: 0.75 (0.62–0.90), *p* = 0.0014	Neutropenia 43.8%(RB) vs. 0.8% (P) Liver related AEs 8.3% (RB) vs. 1.5%(P)
Mayer et al. [73] (PALLAS) (NCT02513394)	III	HoR-positive	Stage II–III	ET+/−palbociclib (2 years)	CDK4/6	5760	iDFS	3-y iDFS 88.2% (PA) vs. 88.5% HR 0.93 (0.76–1.15), *p* = 0.51	72.4% (PA) vs. 14.6%
Loibl et al. [74] (PENELOPE-B) (NCT01864746)	III	HoR-positive	RD after NACT and CPS-EG≥3 or 2 and ypN+	ET/palbociclib vs. ET/placebo (13 cycles)	CDK4/6	1250	iDFS	HR 0.93 (0.74–1.17), *p* = 0.525	79.6% (PA) vs. 20.1% (P)
Chavez-MacGregor et al. [75] (SWOG S1207) (NCT01674140)	III	HoR-positive	-RD and ypN + -pN2 -pN0, T ≥ 2 cm and RS > 25 or MammaPrint high risk -pN1 and RS > 25 or MammaPrint high risk or G3	ET/everolimus vs. ET/placebo (1 year)	mTOR	1939	iDFS	HR 0.94 (0.77–1.14), *p* = 0.52	35% (E) vs. 7% (P)
Tutt et al. [76] (OlympiA) (NCT02032823)	III	HoR-positive and TN (BRCA1-2-mutated)	HoR-positive: ≥pN2 or not pCR and CPS+EG ≥ 3 TN: ≥pN2 or ≥pT2 or not pCR	Olaparib vs. placebo (1 y)	PARP	1836	iDFS	3-y iDFS 87.5% (O) vs. 80.4% (P) HR 0.57 (0.39–0.83), *p* < 0.001 4-y OS 89.8% (O) vs. 86.4% HR 0.68 (0.47–0.97), *p* = 0.009	26.4% (O) vs. 11.7% (P)

Legend: Ph, phase; ET, endocrine therapy; iDSF, invasive disease-free survival; TN, triple-negative; HoR, hormone receptor; G, grading; N, node; OS, overall survival; pCR, pathological complete response; RD, residual disease; RS, recurrence score; T, tumor size; A, abemaciclib; PA, palbociclib; E, everolimus; O, olaparib; P, placebo; RB, ribociclib.

**Table 9 jpm-13-01427-t009:** Ongoing clinical trials exploring targeted therapies in resected breast cancer.

Clinical Trial Identifier (Trial Name)	Design	Target Disease	Number of Patients (N)	Stage	Study Arms	Target	Duration TKIs	Primary Endpoint	Status
NCT04622319 (DESTINY-Breast05)	Phase III, randomized, open-label, 2-arm	HER2-positive	1600	High-risk patients ^1^ with RD after NACT	T-DXd vs. T-DM1	HER2	14 cycles	iDFS	Recruiting
NCT04457596 (CompassHER2 RD)	Phase III, randomized, double blind, 2-arm	HER2-positive	1031	High-risk patients ^2^ with RD after NACT	T-DM1/tucatinib vs. T-DM1	HER2	14 cycles	iDFS	Recruiting
NCT04873362 (ASTEFANIA)	Phase III, randomized, double-blind, 2-arm	HER2-positive	1700	High-risk patients ^3^ with RD after NACT	T-DM1/atezolizumab vs. T-DM1	HER2 + PDL1	14 cycles	iDFS	Recruiting
NCT04595565 (SASCIA)	Phase III, randomized, open-label, parallel group,2-arm	HER2-negative	1200	RD after NACT with high risk of relapse ^4^	sacituzumab govitecan vs. TPC ^5^	TROP2	8 cycles	iDFS	Recruiting
NCT05633654 (ASCENT-5)	Phase III, randomized, open-label, 2-arm	TNBC	1514	RD after NACT	sacituzumab govitecan vs. TPC ^6^	TROP2	8 cycles	iDFS	Recruiting
NCT04752332 (eMonarcHER)	Phase III, randomized, open-label, 2-arm	HoR- and HER2-positive	2450	High-risk disease ^7^	ET/abemaciclib vs. ET	CDK4/6	26 cycles	iDFS	Active, not recruiting
NCT04055493 (ADAPTcycle)	Phase III, randomized, open-label, 2-arm	HoR positive	1670	Intermediate risk according to the ADAPT definition (if missing Oncotype DX clinical intermediate-risk definition)	ET/ribociclib vs. standard-of-care chemotherapy	CDK4/6	26 cycles	iDFS DDFS	Recruiting
NCT04565054 (ADAPTlate)	Phase III, randomized, open-label, 2-arm	HoR positive	1250	High clinical risk ^8^ or intermediate clinical risk: RS > 18 in patients with c/pN 1 or RS > 25 in patients with c/pN 0	ET/abemaciclib vs. ET	CDK4/6	2 years	iDFS	Recruiting
NCT04915755 (ZEST)	Phase III, randomized, double-blind, 2-arm	BRCA-mutated HER2-negative or TNBC	800	I-III with detectable ctDNA following surgery or completion of adjuvant therapy	Niraparib vs. placebo	PARP	3 years	DFS	Prematurely closed ^9^

Legend: NACT, neoadjuvant chemotherapy; iDFS, invasive disease-free survival; DFS, disease-free survival; RD, residual disease; TPC, treatment of physician’s choice; TNBC, triple-negative breast cancer; HoR, hormone receptor; N, node; T, tumor size; G, grade; ET, endocrine therapy; DDFS, distant disease-free survival; RS, recurrence score. ^1^: Defined as inoperable breast cancer at presentation (cT4, N0-3, M0 or cT1-3, N2-3, M0) with RD; or operable at presentation (T1-3, N0-1, M0) with pathological positive node (ypN1-3) after NACT. ^2^: Clinical stage at disease presentation: cT1-4, cN0-3 disease at presentation and RD. Patients with cT1a/bN0 tumors at presentation are not eligible. cN0 eligible if T size ≥ 2.0 cm. cN1-2 eligible if T size ≥ 1.5 cm. ^3^: Clinical stage at disease presentation: cT4/anyN/M0, any cT/N2-3/M0, or cT1-3/N0-1/M0 (participants with cT1mi/T1a/T1b/N0 are not eligible). ^4:^ high risk of recurrence defined by either: for HoR-negative: any residual invasive disease > ypT1mi; for HoR-positive disease: a CPS + EG score ≥ 3 or CPS+EG score 2 and ypN+. ^5^: Capecitabine or platinum-based chemotherapy or observation. ^6^: Pembrolizumab or pembrolizumab + capecitabine. ^7^: After NACT: ypN1, or T ≥ 5 cm, or a residual tumor of any size that has direct extension to the chest wall and/or skin (ulceration or skin nodules). If surgery upfront: pN2 or pN1, and G3 or T ≥ 5 cm. ^8^: Criteria for high clinical risk included high risk by PROSIGNA^®^ (score > 60 in N 0 and >40 in N+) or EPclin^®^ (score > 3.3287), or MammaPrint^®^ within clinical routine. ^9^: The ZEST trial permanently stopped enrollment (less than 5% of the 800 planned patients had been randomized).

## Data Availability

Not applicable.

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
