# Peer review of "Adjuvant Targeted Therapy in Solid Cancers: Pioneers and New Glories"

_jpm, 2023, doi:10.3390/jpm13101427_

Round 1
Reviewer 1 Report
Article title: Adjuvant Targeted Therapy in Solid Cancers: Pioneers and New Glories
The authors here described in-depth the recent progress in targeted therapy concerning early-stage gastrointestinal stromal tumors (GIST), melanoma, non-small cell lung cancer (NSCLC), and breast cancer. The review is well structured, and descriptive and provides detailed information about the concluded as well as ongoing clinical trials. However, the following minor comments need to be considered for a better understanding of the manuscript.
1. The authors can change the title of the study in Table 1B to clinical trial.gov Identifier. In Table 1A in the trial column, the clinical trial identifier is not mentioned. It is better to mention the Identifier details wherever necessary. It would be good if the authors also mentioned the targets for the drug. For example, Dabrafenib-Trametinib in BRAF + MEK inhibitor.
2. Figure 2 mentions T-DM1 for breast cancer treatment. Please provide details for T-DM1 (Ado-trastuzumab emtansine).
3. It is better if authors also provide some resistance mechanisms and alternative treatments for a few of the inhibitors in use. For example, the resistance mechanism of BRAF + MEK Inhibitor in melanoma leads to mutation in pathways such as PI3K pathway, NRAS mutation, etc. This approach can be using a combination such that the drug can inhibit both the MAPK and PI3K–Akt pathways.
4. The concluding remarks can be improvised with more details about new therapies.
Author Response
Dear Reviewer 1,
Please find enclosed the revised manuscript entitled “Adjuvant targeted therapy in solid cancers: pioneers and new glories”.
We would like to express our gratitude to the reviewers for their positive feedback and also extend our thanks for their punctual comments, which have guided us in understanding the aspects that need improvement in our manuscript.
We marked the changes using a different color (yellow) of text (in the manuscript file). We updated also the tables and the figures.
All the revisions suggested were performed as a point-by-point rebuttal, as follows:
Reviewer #1:
The authors here described in-depth the recent progress in targeted therapy concerning early-stage gastrointestinal stromal tumors (GIST), melanoma, non-small cell lung cancer (NSCLC), and breast cancer. The review is well structured, and descriptive and provides detailed information about the concluded as well as ongoing clinical trials.
Response: Thank you for appreciating our manuscript.
1 - The authors can change the title of the study in Table 1B to clinical trial.gov Identifier. In Table 1A in the trial column, the clinical trial identifier is not mentioned. It is better to mention the Identifier details wherever necessary. It would be good if the authors also mentioned the targets for the drug. For example, Dabrafenib-Trametinib in BRAF + MEK inhibitor.
Response: Thank you for the comment; we changed the titles and provided clinicaltrial.gov identifier for each study cited in the tables. We also specified the molecular targets of reported therapies.
2 - Figure 2 mentions T-DM1 for breast cancer treatment. Please provide details for T-DM1 (Ado-trastuzumab emtansine)
Response: Thank you for the comment; we specified details in the text [lines 330-331] and reported full name in the figure 2.
3 - It is better if authors also provide some resistance mechanisms and alternative treatments for a few of the inhibitors in use. For example, the resistance mechanism of BRAF + MEK Inhibitor in melanoma leads to mutation in pathways such as PI3K pathway, NRAS mutation, etc. This approach can be using a combination such that the drug can inhibit both the MAPK and PI3K–Akt pathways.
Response: Thank you for the comment, we added details in the discussion paragraph.
4 - The concluding remarks can be improvised with more details about new therapies.
Response: Thank you for the comment; we expanded the Conclusion paragraph.
Reviewer 2 Report
Sposito et al. wrote a good review summarizing the current progress of adjuvant targeted therapy. The authors did a great job on putting the available adjuvant target therapy studies and clinical trials together. The organization of the manuscript is good. The studies and clinical trials were divided into four groups based on the tumor types they are targeting. This makes the manuscript easier to follow. However, the manuscript is wordy and lacking figures. In addition to the results of the clinical trials, readers cannot get more interpretation/meaning of these results after reading the manuscript. Review should not only be a collection of results but also needs to have interpretations/thoughts from the authors. The following issues may be resolved to improve the presentation of this review.
Minor issues:
1. Abbreviations were used before the full names were introduced. For example: “CI. In addition, there are too many abbreviations in the manuscript which makes it hard for readers to follow through the manuscript. Reducing the usage of unnecessary abbreviations will make it more friendly to general readers.
2. Line 109: “1.5 cases/100.000/year” should be “1.5 cases/100,000/year”
3. Throughout the paper, the parameters of clinical trials were used frequently. For example, RFS, OS, HR, CI, pCR etc. The authors should first explain what the parameters are, what do the values mean, and what values would be considered effective.
Major issues
1. The manuscript is well-organized. However, the overall manuscript is wordy and needs more figure demonstration. The authors describe the setup and results of different studies and clinical trials without much interpretation/explanation. The authors can make figures to summarize the results of the clinical trials for each type of tumor. For example, there were a few clinical trials for “Melanoma” and the DFS, OS, HR values were reported for each trial. Figures can then be made for all the values and all trials. The values of each parameter that are considered effective can be added as a line to each figure to help judge whether the results from the clinical trials reach the level and whether the tested adjuvant targeted therapy can be considered effective.
Author Response
Dear Reviewer 2,
Please find enclosed the revised manuscript entitled “Adjuvant targeted therapy in solid cancers: pioneers and new glories”.
We would like to express our gratitude to the reviewers for their positive feedback and also extend our thanks for their punctual comments, which have guided us in understanding the aspects that need improvement in our manuscript.
We marked the changes using a different color (yellow) of text (in the manuscript file). We updated also the tables and the figures.
All the revisions suggested were performed as a point-by-point rebuttal, as follows:
Reviewer #2:
The authors did a great job on putting the available adjuvant target therapy studies and clinical trials together. The organization of the manuscript is good. The studies and clinical trials were divided into four groups based on the tumor types they are targeting. This makes the manuscript easier to follow.
Response: Thank you for appreciating our manuscript.
Minor issues:
1 - Abbreviations were used before the full names were introduced. For example: “CI. In addition, there are too many abbreviations in the manuscript which makes it hard for readers to follow through the manuscript. Reducing the usage of unnecessary abbreviations will make it more friendly to general readers.
Response: Thank you for the comment; we reduced the number of acronyms and added explanations.
2 - Line 109: “1.5 cases/100.000/year” should be “1.5 cases/100,000/year”
Response: Thank you for the comment; we corrected the typo.
3 - Throughout the paper, the parameters of clinical trials were used frequently. For example, RFS, OS, HR, CI, pCR etc. The authors should first explain what the parameters are, what do the values mean, and what values would be considered effective.
Response: Thank you for the comment; we added an explanation at the beginning.
Major issues
- The manuscript is well-organized. However, the overall manuscript is wordy and needs more figure demonstration. The authors describe the setup and results of different studies and clinical trials without much interpretation/explanation. The authors can make figures to summarize the results of the clinical trials for each type of tumor. For example, there were a few clinical trials for “Melanoma” and the DFS, OS, HR values were reported for each trial. Figures can then be made for all the values and all trials. The values of each parameter that are considered effective can be added as a line to each figure to help judge whether the results from the clinical trials reach the level and whether the tested adjuvant targeted therapy can be considered effective.
Response: Thank you for the comment. Considering the nature of our manuscript (a narrative review), we decided to include all the available clinical trials and their main results in the tables. We agree with the reviewer’s comment, so we have added to Figure 1 the results (HR and CI of the main endpoint) of the main trials leading to drug approval in that specific disease.